# A Review on Human–AI Interaction in Machine Learning and Insights for Medical Applications

**DOI:** 10.3390/ijerph18042121

**Published:** 2021-02-22

**Authors:** Mansoureh Maadi, Hadi Akbarzadeh Khorshidi, Uwe Aickelin

**Affiliations:** School of Computing and Information Systems, The University of Melbourne, Melbourne 3010, Australia; hadi.khorshidi@unimelb.edu.au (H.A.K.); uwe.aickelin@unimelb.edu.au (U.A.)

**Keywords:** human-in-the-loop machine learning, human–AI interaction, interactive machine learning, medical applications

## Abstract

Objective: To provide a human–Artificial Intelligence (AI) interaction review for Machine Learning (ML) applications to inform how to best combine both human domain expertise and computational power of ML methods. The review focuses on the medical field, as the medical ML application literature highlights a special necessity of medical experts collaborating with ML approaches. Methods: A scoping literature review is performed on Scopus and Google Scholar using the terms “human in the loop”, “human in the loop machine learning”, and “interactive machine learning”. Peer-reviewed papers published from 2015 to 2020 are included in our review. Results: We design four questions to investigate and describe human–AI interaction in ML applications. These questions are “Why should humans be in the loop?”, “Where does human–AI interaction occur in the ML processes?”, “Who are the humans in the loop?”, and “How do humans interact with ML in Human-In-the-Loop ML (HILML)?”. To answer the first question, we describe three main reasons regarding the importance of human involvement in ML applications. To address the second question, human–AI interaction is investigated in three main algorithmic stages: 1. data producing and pre-processing; 2. ML modelling; and 3. ML evaluation and refinement. The importance of the expertise level of the humans in human–AI interaction is described to answer the third question. The number of human interactions in HILML is grouped into three categories to address the fourth question. We conclude the paper by offering a discussion on open opportunities for future research in HILML.

## 1. Introduction

Recent impressive developments of Machine Learning (ML) methods have created many star applications in various fields of science and industry. Briefly, ML methods as a part of Artificial Intelligence (AI) are tools that automatically learn from sample data (training data) and provide insightful knowledge. Different types of ML methods can be categorised into supervised, unsupervised and semi-supervised ML methods. Supervised ML methods use training data with labels and assign labels to all feasible inputs. Unsupervised ML methods use a training data set without labels and group data by finding similarities among training data. Semi-supervised ML methods use a training data set consisting of labelled data and unlabelled data (mostly unlabelled data) and are the combination of supervised and unsupervised ML methods. The recent trend of using ML methods to deal with real-world problems has made us ask: Are these automatic approaches enough or do we need human involvement? Although several powerful ML methods have been presented and implemented to solve real-world problems, it has been suggested that ML methods are not enough in critical decision making approaches (such as medical applications) and humans should be involved in the ML loop [1]. We believe that for many future ML applications, despite the significant improvement in designing ML methods, the collaboration of humans and ML methods cannot be denied because two heads are better than one [2]. These two heads are ML methods and humans. For example, in Nascimento et al. [3], the combination of software engineers of different levels of expertise as human experts and ML methods was investigated and compared for a case study on streetlight automation planning. The results showed that human experts outperform the ML methods in some experimental conditions, but the ML methods have better performance in other conditions. So, human–AI interaction was introduced as the solution for better streetlight automation planning, as it included both human and ML methods capabilities.

Applications of ML methods to deal with real-world problems have revealed some drawbacks that result in unsuccessful outcomes and a necessity for improvement. ML methods need the availability of high-quality, unbiased and inclusive training data for high performance. In many cases, a lack of sufficient high-quality data usually results in unsatisfactory outputs. For example, in Roccetti et al. [4], neural networks were trained on a huge database of water consumption measurements and were unsuccessful to predict test data with sufficient precision. While the ML methods used could only approximate the relationship between model inputs and model outputs. Therefore, the accuracy of these methods was confined [5]. Another example in Weber et al. [6], found that an automatic image inpainting process by neural networks could not result in accurate and satisfactory outputs and only human knowledge could improve it. In addition, the difficulty of interpreting internal ML rules is mentioned as another drawback.

Humans are the agents that can diagnose why and how ML methods are not successful. Human interaction helps to reveal the drawbacks of ML methods and also satisfies the humans who are consumers of the ML methods. Also, the ability of humans using abstract thinking, as well as a large collection of past experiences and knowledge, cannot be denied. Especially when it comes to inducing new patterns and complicated processes. Although ML methods are able to provide well-expressed and exact responses to well-structured problems, they are not so good at solving vague and ill-posed problems [7]. So, the collaboration of humans with ML methods can result in more successful and impressive applications of the ML for these problems. To exemplify, in Yang et al. [8], at first, only automatic data processing was performed for text classification using ML methods. Then if the results didn’t have enough quality in comparison to a predetermined threshold, a human-in-the-loop approach was presented until the desired results were received.

For these reasons, ML needs methods that engage humans directly into the ML process and adapt the ML process according to a user’s preference. To satisfy this need, Human-In-the-Loop ML (HILML) and Interactive Machine Learning (IML) have recently emerged as two common terms that indicate collaboration between humans and ML methods. These two methods slightly differ in terms of defining collaboration scenarios between humans and ML methods. Definitions, concepts and differences of these two terms will be described in the next section.

Future success of human–AI interaction in ML applications, especially in high-risk decision-making areas like medical applications, requires a comprehensive review for practitioners and researchers to realise the importance of domain human experts’ interaction in ML applications. In this paper, we provide important and necessary information for human–AI interaction in ML applications by providing four research questions and discussing them as the result of this review in separate sections. “Why should humans be in the loop?”, “Where does human–AI interaction occur in the ML processes?”, “Who are the humans in the loop?” and “How do humans interact with ML in HILML?” are questions that by answering them, can guide researchers to comprehend strengths, weaknesses and application scenarios of human–AI interaction in ML applications. This paper will focus on the human–AI interaction challenges and future research opportunities in medical ML applications that are presented by these questions.

### 1.1. Human-in-the-Loop Machine Learning: Definition and Terminology

HILML enables users to change the results of ML methods by incorporating human skills and expertise. This allows ML methods to learn tasks that cannot be achieved solely, cannot deal with unknown situations and cannot be adapted to environmental dynamics [9]. In the literature of HILML, human interaction is crucial in ML applications for creating training data, learning process and model refinement [9,10]. Although sometimes in ML applications, HILML and IML have been used as the same terms for human–AI interaction. However, it can be said that IML focuses more on the training process and model refinement in ML, so that humans could steer ML methods to get desired results. While in HILML, humans can interact in all steps of the ML process including data and model for better performance of the ML methods. So HILML is a more general term than IML for human–AI interaction, which has the concept of IML in its definition [11].

### 1.2. The Difference between This Survey and Former Ones

The first attempt to involve humans in ML applications was presented in the paper of Ware et al. [12]. In this paper, generating classifiers by domain human experts was introduced as an IML method. Then, Fails and Olsen [13], intending to refine an ML model in image segmentation system, proposed an IML approach which consisted of an iterative train-evaluate-refine life cycle. The collaboration between humans and ML methods (IML and HILML) have been investigated in several survey papers. However, there are only few review papers that shape this area and review current work progress. A 15-step “data understanding pipeline” as the main steps of IML was outlined in Aodha et al. [14] in 2014 for scientists to answer their research questions, where the questions strongly depend on data collection and data modelling. In this human–AI collaboration workflow, “hypothesis formation” is the first step and “publicising of results” is the last one. The proposed workflow is comprehensive; however, it includes steps that do not satisfy the real collaboration between scientists and ML methods [15].

The role of humans in IML was investigated in Amershi et al. [16] using several case studies. “Image segmentation” and “Gesture-based music” were initially used as two case studies to exemplify the IML process and demonstrate the power of IML as a tool for end-users. Then, to study human interaction in IML, some case studies were investigated to show how humans may quash assumptions of the ML, how humans want to interact with ML methods in richer ways, and how transparency in ML methods improves user experience in IML. In this survey the key benefits of IML were introduced as studying the users of the learning systems; no limitation in the interaction of humans and ML methods; and the necessity of studying ML methods and users together. Although this paper investigates the initial requirements and necessity of better user and ML interaction in IML, it does not cover the role of humans in different stages of IML methods.

Dudley et al. [15] proposed a workflow focused on key interaction stages in IML including six activities as “feature selection”, “model selection”, “model steering”, “quality assessment”, “termination assessment” and “transfer”. In their paper, four key components of the IML models were introduced as data, model, user and interface, where the main characteristics of the interface elements in an IML model (as well as interface design challenges) were investigated thoroughly. Also, a survey of IML applications according to the data type of IML methods was conducted to highlight the requirements of developing IML interfaces. These IML methods consisted of “text”, “images”, “time series data”, “assisted processing of structured information” and “raw numerical data”. Their paper focused more on presenting an informative and comprehensive review on designing user interface in IML, while extended an IML workflow to investigate the role of humans in ML applications. However, in the presented IML workflow, human collaboration in data pre-processing is not considered as a step for human–AI collaboration, while it is a key step in human–AI interaction specially when ML methods are unsupervised.

The collaboration between humans and ML methods is also investigated in the paper of Wang et al. [17]. The focus of their paper was in presenting a technical framework of crowd-assisted (not necessarily experts) ML methods which consisted of three main stages: “Data pre-processing”, “Feature discovery and learning” and “Model assessment and refinement”. In their paper, the role of crowd workers to compensate the drawbacks of ML methods was categorised into “crowdsourcing for labelling”, “crowdsourcing for data cleaning”, “crowdsourcing for discovering features with limited data”, and “crowdsourcing for model assessment, explanation and refinement”. Their paper only highlights the role of human crowd workers in collaboration with ML methods, not humans in general term.

In the field of medical applications, some papers related to human behaviours in human-Al interaction have been presented. A domain-expert-centric process model for knowledge discovery was proposed in Girardi et al. [18] in 2015, that showed how a domain expert can collaborate with highly complicated ML tasks in knowledge discovery. For expert driven knowledge discovery model, six steps of “data modelling”, “data acquisition”, “data validation”, “data preparation”, “data analysis” and “evaluation” were suggested as the key stages of knowledge discovery in biomedical research. Although this process model is valuable in biomedical knowledge discovery, it does not explain collaboration between human and ML adequately [15].

Another research project that studied the challenges and benefits of IML in health informatics is the paper of Holzinger in 2016 [19]. In their paper, a brief general description about the difference between IML and ML and the importance of human–AI interaction in ML applications were presented at first. Then, three examples of IML application as “subspace clustering”, “protein folding” and “k-anonymization of patient data” were described. Preference learning, reinforcement learning, active learning and active process learning were briefly discussed and introduced as the basis for IML. Although their paper investigates three important health informatics examples to show how human and AI collaborate, it does not present a general framework for human–AI interaction applied in medical ML.

Regarding previous research in human–AI interaction in ML applications, we conclude that there is no in-depth survey to cover current efforts of human collaboration for different ML applications, including medical applications. In our survey, we firstly provide a comprehensive summary of current research on human–AI interaction to identify the role of humans in collaboration with ML methods regarding all steps of the ML process. Then, we highlight the role of humans in collaboration with ML methods in medical applications to identify challenges and future directions in this area.

### 1.3. Contributions of This Survey

The first aim of this paper is presenting a generalised and comprehensive overview of human–AI interaction. We focus on HILML as a more general key term for human–AI interaction to review ML applications and present a framework for human–AI collaboration. This framework is organised according to the three main stages of ML models: data producing and pre-processing, ML modelling and ML evaluation, and refinement (see Figure 1, Figure 2 and Figure 3). In this framework, corresponding human interactions are adopted in each stage to improve the performance of ML methods. It provides a perspective that researchers can use to quickly understand different aspects of human–AI interaction. As well as presenting a comprehensive framework, we try to review recently published papers to survey human–AI interaction for current real-world ML applications. We also investigate the importance of the human role in HILML, as well as the characteristics of humans who collaborate with ML methods and categorise human issues in HILML by discussing the research questions. Moreover, we highlight the role of humans in medical ML applications and review important human–AI collaboration approaches in this area to investigate research challenges, research gaps and future research. To summarize, the key contributions of this paper are threefold in that we conduct a comprehensive review on human roles in human–AI interaction in ML applications, we provide an overview for the current research in human–AI interaction area, and we discuss the role of humans in collaboration with ML methods in medical applications and identify challenges and future directions.

## 2. Materials and Methods

### Survey Methodology

To collect relevant papers for this survey, “human-in-the-loop machine learning”, “human in the loop” and “interactive machine learning” were three major keywords that we used to search papers indexed with Scopus and Google Scholar. We also screened the most important and related conferences in the field of ML applications which was cited in IEEE and ACM websites to find high-quality work. We reviewed all the recent (2015–2020) highly cited papers to describe human–AI interaction in current ML applications, and highlighted recent work in medical HILML applications. However, we considered and reviewed selected highly cited and important papers published as early as 2001 to not lose any vital information about human–AI interaction in ML applications.

## 3. Results

To discuss review results, we designed four research questions to cover all concerns related to HILML. In each section, we answered one research question. In the next section, the reasons behind human involvement in ML applications is presented based on motivation provided in different papers related to HILML.

In Section 3.2, the collaboration between humans and ML methods regarding different steps of the ML process is categorised and described.

The characteristics of humans in HILML in terms of the level of expertise is described according to the recently published papers in Section 3.3.

Section 3.4 gives some perspectives about the types of human–AI interaction in terms of the number of human involvements in HILML.

In Section 3.5, human–AI interaction in medical ML applications is investigated and classified according to the human–AI interaction steps explained in Section 3.2. Also, in this section, a discussion on new research challenges and future research opportunities is conducted.

### 3.1. Why Should Humans Be in the Loop?

The necessity of human–AI interaction in ML applications and the results of this collaboration have been investigated in several papers. In these papers, different reasons have been mentioned to highlight the importance of human–AI interaction. In this section, we identify and categorise three main reasons for human–AI interaction in ML applications. In the following sections, we describe these categories in detail.

#### 3.1.1. Tasks Are Too Complicated

When it comes to the application of ML methods to address complicated real-world problems, the results of using ML methods are not always satisfactory. A solution to improve the results of these problems is the human-in-the-loop approach. For example, Kunneman et al. [20], applied Support Vector Machine (SVM) and naïve Bayes on TwiNL (a data set that includes Dutch Twitter messages) to monitor and detect negative stances about vaccination in Twitter messages. Their best results with AUC of 0.66 and F1-score of 0.57 were achieved via SVM. However, they mentioned that using only ML methods to measure stances is not enough for detection and a human-in-the-loop approach was suggested for better results in this complicated problem. Cadavid et al. [21], by presenting a systematic review paper on ML aided Production Planning and Control (ML-PPC), asserted that 75% of the possible research areas in this field are almost not addressed well. Ignoring human–AI interaction when ML methods are implemented in PPC was introduced as a reason for this issue. Portelli [7], by reviewing the efficiency of the ML methods to deal with heterogeneous, massive and dynamic ecological datasets, noted that the contribution of humans in understanding pattern, processes and relationships should not be neglected. She reviewed the tensions between ML methods and human interpretations and concluded that the humans should have remained in the design of landscape ecology applications when ML methods are implemented.

A common discussion in ML applications with humans in the loop is that people can provide knowledge about the complicated tasks that cannot be captured by the data sets [22]. In some applications of ML, tasks are highly complicated and the ML methods are unable to efficiently cope with the problem, so the involvement and collaboration of humans can help. Sakata et al. in [23] used human collaboration to provide feature values for their proposed ML approach because the definition of the features of the problem was abstract and consequently too difficult to extract automatically. Bahrami and Chen [24] proposed human-in-the-loop based neural networks using crowd-generated inputs to produce unified web APIs (Application Programming Interfaces). In their paper, it is reported that using only ML methods to produce APIs (like SVM and deep learning methods) may not be sufficiently accurate as any incorrect extracted information causes a problem in the software life-cycle. So, an additional human involvement is required to produce APIs to adjust incorrect information produced by ML. Roccetti et al. [4] used human experts’ knowledge by defining some semantics to select high-quality instances from the data set, to train neural networks on fifteen million water meter readings with the goal of increasing the accuracy of the ML prediction.

Several papers for various ML application areas have been presented to improve the quality of ML methods where the role of humans to compensate for the deficiencies of ML methods is highlighted. Weber et al. [6] proposed an Interactive Deep Image Prior (IDIP) which was based on Deep Neural Network and human-in-the-loop for image restoration. The role of humans was refining the output of the IDIP iteratively by providing some guidance specially missing semantics and controlling IDIP for better restoration as well as dealing with overfitting issue. The results on the Mongao Grottoes data set showed the superior performance of IDIP over DIP in terms of Local Mean Square Error (LMSE) and Dissimilar structural similarity Index Measure (DSSIM).

In Feder et al. [25], using three deep neural networks named CNN, BiLSTM and Bert-based attention model to detect demographic traits in clinical notes, a bootstrapped active learning iterative process was designed in which humans relabelled some instances manually in the training data set to retrain classifiers to improve the prediction accuracy. Results on the MIMIC-III data set showed the outperformance of the human-in-the-loop classifiers rather than traditional classifiers of Random Forest (RF), naïve Bayes and logistic regression in terms of recall, precision and F1-score. In Wen et al. [26], a HILML method was presented for the problem of tracking a moving target in the video stream where the presented ML methods in this field had limitations to track targets and recover tracking failures. In their paper, using a human expert in tacking loop improved the performance of the ML method so that the output results showed the significant difference between the performance of two practical ML methods and the proposed HILML approach.

#### 3.1.2. ML Methods Are Not Transparent and Explicable

Interpretability and explicability contradict with ML methods that usually only aim for the correctness of results. While researchers try to develop ever more complicated ML models with higher abilities to correctly predict, the problem is that these models often work as black boxes and users cannot trust them as they do not know the reasons behind their decisions. For example, in drug discovery, the lack of interpretability is mentioned as a reason for ignoring ML methods, while making decisions only by humans leads to biased and low-quality results [27]. Two categories of glass box and black box ML methods have been presented in the literature. In the glass box ML methods, the inner working of the methods is easy to understand but in complicated tasks, these methods are not successful to achieve high-quality results. However, the performance of this kind of ML methods can be improved by involving human experts in the loop to cope with real-world problems. For example, in [28], some learning rules were changed or added to the rules extracted from a glass box method by humans with the aim of generalization of the ML methods to deal with the problem of overfitting in data mining. In contrast to glass box ML methods, black box methods are more powerful in prediction while the results are not explainable or interpretable [29]. Neural Networks (NNs) and RF are methods that can be listed in this category. An effective approach that can simultaneously satisfy the need to have powerful ML methods to cover the complexity of real-world applications and increase the interpretability to keep the process understandable is HILML. For example, In Stroganov et al. [27], it was claimed that the human-in-the-loop approach improves the transparency of the ML methods to be used in the drug discovery field.

To improve model interpretation, some quantifiable approaches are presented in the HILML literature which are related to the association of predictions and responses. One of these approaches is calculating the model’s feature importance, so users understand how features lead to the results [11,30]. In some applications, defining features by users can increase ML transparency. Cheng et al. [31] gave the power of generating features to human crowd workers to generate understandable features that can improve the human judgment and subsequently ML transparency. Correia and Lecue [32] presented a human-in-the-loop feature selection approach that renders the ML decision making more interpretable for human users. Drobnic et al. [29] presented a feature selection approach with the help of human experts to improve RF’s interpretability. In [33], Kulesza et al. presented a naïve Bayes learning system in text classification so that human experts were able to specify how much a feature matter in the classification with the purpose of increasing ML transparency.

In addition to human collaboration in feature issues as the most common collaboration area to increase ML transparency in HILML, humans have been involved in different steps of ML applications to evaluate and refine ML results in terms of transparency and explicability [16]. For example, parameter tuning, selecting instances from the data sets for training to get desired results, adding some samples to the training data set and labelling and relabelling some data instances are human activities which result in steering and refining ML methods. They can help to increase transparency and explicability in ML applications. In Section 3.2, we describe these kinds of human–AI interaction in ML applications.

#### 3.1.3. ML Results Are Not Satisfactory for Humans

The results of ML methods are not always acceptable subjectively for humans. In fact, human needs and understanding conflict with ML method results for some situations. So it is the role of humans to control the ML process to get more desirable results. For example, in Weber et al. [6], humans were more satisfied subjectively with the quality of the restored images generated by IDIP (human-in-the-loop approach) than DIP. This was tested using questionnaires and different groups of human experts. In [34], Yuksel et al. involved humans to refine the ML outputs which were video descriptions for visually impaired users. Results of this collaboration showed more user’s satisfaction from the human-in-the-loop approach than only the ML approach measured subjectively. Boddy et al. [35] introduced their proposed HILML method as a model that the human experts can include their subjective assessment of malicious intent to identify security incidents when electronic patient records are used for analysis. In their paper, to preserve privacy, although the ML outlier detection methods were used to identify the security accident when electronic patient record data were used, the outliers were assessed by a human expert because maliciousness is not particularly an event rarity as an outlier. In [36], Zhang et al. asserted that in the analysis of large qualitative open-ended survey data, there are some areas of incongruity between ML methods and human analysts and HILML was suggested to receive the feedback from humans to achieve satisfactory results.

### 3.2. Where Does Human–AI Interaction Occur in the ML Processes?

In this section, we discuss the tasks associated with the collaboration of humans and ML methods. As we explained in the introduction, humans collaborate with ML methods in the ML processes from data producing to model evaluation and refinement. In this section, we investigate where humans can be in the loop and classify the human–AI interaction processes into three general categories of human–AI interaction in data producing and pre-processing, human–AI interaction in ML modelling, and human–AI interaction in ML evaluation and refinement. In each category, we discuss the human roles in human–AI interaction which have resulted in better performance of ML methods. Also, we review state-of-the-art HILML application papers to provide an overview for researchers about new research areas of HILML applications and the role of humans to improve ML performance.

#### 3.2.1. Human-in-the-Loop for Data Producing and Pre-Processing

The success of ML methods in real-world applications is dependent on not only the design of the ML procedures but also the quality and semantics of the data used for training ML methods [4]. Data quality has an important role in the success of ML approaches in prediction. The higher the quality of the data set is, the more accurate results will be achieved. A good example of the importance of data quality, as well as human–AI interaction, to improve ML performance can be seen in Roccetti et al. [4]. In their paper, firstly, neural networks were trained on a big data set without considering the quality of the data set and the algorithms were unable to predict correctly regarding a determined precision. Secondly, the neural networks were trained on some samples of the data set as a new data set, in which most impurities of the main data set were filtered out by data cleaning methods. The results showed the accuracy of the prediction increased, while the new data set was statistically different and smaller than the main data set. Finally, in the third step, the neural networks were trained on a new data set from the main data set acquired by the cooperation of human experts and data cleaning methods. Human experts defined new data semantics to determine a representative form of the data to be considered in the training data set. In this case, a higher standard of data quality achieved without violating the underlying statistical characteristics of the main dataset resulted in excellent accuracy of ML prediction.

We discuss human tasks in human–AI interaction for data producing and pre-processing in ML separately in two subsections. In each subsection, for each human–AI interaction task, we describe the role of humans, the aim of the human collaboration and the results of this collaboration by providing indicative case studies related to HILML. To present a general overview of how humans interact in data producing and pre-processing and how this interaction address ML challenges in this stage, we provide a framework that is shown in Figure 1.

##### Data Producing

The role of humans in producing data before and during the ML process in HILML applications can be classified into two categories of providing data set samples and data labelling. In the following sections, these two categories are described in detail.

Providing data set samples

Generating data from humans provides a convenient, reliable and efficient source for training ML methods [23,37]. Producing data by crowd workers and domain human experts are two common strategies for making data set samples in HILML applications. In this area, researchers have focused more on comparing the quality of the data samples generated by different groups of humans and evaluating the performance of the ML models derived by these data instances.

Producing data set samples can be performed by crowdsourcing. For example, Sakata et al. [23] implemented crowdsourcing to generate a training data set in the problem of recognising painters from the paintings. In their paper, using some questions about the characteristics of the painting, the features’ values were determined by human crowd workers. When crowdsourcing platforms are used to produce data, although in some cases these platforms are successful to generate data efficiently, a compulsive limitation is imposed to the problem that the produced data can be highly error-prone [37]. Different approaches have been presented in the crowdsourcing literature to deal with this problem, such as finding new incentives for crowd workers to work more efficiently [38], collecting and aggregating numerous crowd workers’ responses to have reliable data [39], considering the skills of the crowd workers [40] and so forth. Based on the literature, it can be deduced that the more the level of domain expertise is, the more the probability of producing more accurate and high-quality data sets is.

Despite factors like cost and time that can affect the domain human expert involvement, the role of domain human experts to produce high-quality instances is highlighted in the HILML literature. For example, we can refer to the paper [41] that Huang et al. presented a HILML approach in which domain human experts were responsible to generate training instances. In their paper, the training instances were rated by medical experts through a scoring strategy for identifying benign and malignant breast tumours in an ultrasound. The successful performance of the proposed ML approach verified the power of the ML approach, as well as the quality of the training data set produced by experts. For more investigation, the performance of the proposed ML method regarding two kinds of training data sets produced by two groups of clinicians, with different levels of expertise as experts and interns, was tested. The results showed that in terms of sensitivity, specificity and accuracy, the proposed ML had better performance using the experts’ generated training data than intern ones. Another example of human–AI interaction to produce training data set can be seen in recommender systems as a HILML approach that focusing on data capturing by interpreting humans’ actions when they are interacting with an algorithm [15]. So in ML-based recommender systems, humans have the role of generating data sets for ML approaches. The aim of designing recommender systems is creating software which adapts to the humans’ requirements.

In addition to the role of humans to produce data for training ML models, humans are responsible to provide some instances to refine the outputs of the trained models. For example, Liu et al. [22] proposed a HILML approach in an unknown unknown detection problem to detect misclassified instances with high confidence and improve the performance of the classifier by adding some labelled instances to the training data set.

Data labelling

Data labelling is the most common cooperation of humans with ML methods in applications. Tagging data and providing labels for them is important especially in training supervised ML methods. The role of humans to label data as well as the related issues arises in this cooperation such as the quality of the labels provided by humans, the number of human experts collaborating in data labelling and the level of the expertise for humans who are responsible for data labelling have been investigated in several studies. Like the data producing step, two common strategies for human collaboration in data labelling have been presented as providing labels for all samples of the data set and providing labels for some samples of the training data set. Also, crowd workers and domain human experts are two commonly used key terms that indicate the role of humans in human–AI interaction for data labelling.

Human collaboration to provide data labels for the whole data set is highlighted in supervised ML methods. The main research question related to this area is how do humans provide data labels accurately with minimum cost? The performance of different groups of humans to address this question has been investigated in the literature. For example, in [42], Michalopoulos et al. investigated data labelling for docket entry data classification via lawyers and described the labelling process to produce accurate data labels. In [43], two structures for data labelling in order to help people to define and refine their concepts in data labelling are proposed and compared. Chang et al. [44] proposed a collaborative crowdsourcing approach for data labelling in ML applications. In their paper, a guideline was presented to support the ambiguous items in data labelling in three stages as vote, explain and categorise. In Zhou et al. [45], providing data labels in image processing for plant phenomics by two crowdsourcing platforms and one group of students were investigated and compared in terms of accuracy, speed and time. The results showed that crowdsourcing platforms were more successful than students in data labelling. However, it is noted that the domain experts’ data labelling is more accurate than crowdsourcing and humans without the high level of domain expertise. Snow et al. [46] compared the quality of labels provided by crowdsourcing with domain experts in natural language processing and got the conclusion that for a recognition task in natural language processing, the performance of 4 non-experts to generate high-quality data labels is almost equal to one domain expert.

Providing data labels for the whole data set by only humans is a hard, time consuming and tiresome task [47]. So different algorithms and ML methods have been presented to improve data tagging process in which the advantages of humans and their cognitions cannot be denied. Labelling and relabelling some samples in the data set are tasks that humans do, intending to guide the ML methods to get better results during the ML training process especially in semi-supervised ML. In Wen et al. [26], humans were responsible to provide some labels for the data at the first step of the ML method by depicting a bounding rectangle around the moving target in a video stream. The role of humans was monitoring the tracking process and relabelling if the ML method was unsuccessful to track the target. In [35], in a semi-supervised ML approach, Boddy et al. provided labels regularly for some data samples so that the ability of the presented method was improved in outlier detection using human experts’ feedback. In [48], Wrede and Hellander proposed a human-in-the-loop semi-supervised ML method to explore high-dimensional stochastic models of gene regulatory networks that the role of humans was data labelling to inform the proposed method about the more interesting behaviours.

The role of human experts for data labelling in ML applications is highlighted in active learning. As obtaining data labels in ML applications is usually expensive, human experts label selective samples in active learning. Different studies to use active learning in ML applications have been presented, in which the role of humans is labelling selected samples. For example, Feder et al. [25] employed humans to relabel a very small fraction of training data which was wrongly classified by classifiers in their proposed active learning approach to detect demographic traits in clinical notes. In [47], a human-in-the-loop active learning method was presented based on brain-computer interface and deep learning to tag target visual data during the training process. In the proposed method, users were connected to the electroencephalograms electrodes and images include target images and non-target images were shown to them as a rapid serial visual presentation. Based on the users’ brain activity when target objects were shown, the computer learned to identify the target object. In [49], an active learning approach was presented for wireless IoT intrusion detection. In [50], active learning was applied in the annotation of aerial images in environmental surveys.

##### Data Pre-Processing

Data pre-processing or data cleaning as one of the main stages in ML approaches is related to detecting and fixing errors of the data set. To deal with outliers, inconsistent data, missing values, noisy data, duplicates and other errors in data sets that affect the quality of the data and ML methods, a plethora of algorithms and methods have been presented. Although many of these methods are based on the power of computers and algorithms to clean the data, it can be stated that almost all data cleaning methods need human collaboration specially for describing data quality rules and recognising and fixing errors [51]. In recent papers, human cooperation at different stages of data cleaning for validating repairs results, providing information, rules and knowledge and resolving ambiguity is advantageous [52].

Generally, data pre-processing approaches are classified into two categories. The first category is related to introducing new data pre-processing algorithms for specific kinds of data issues and the other one is developing ML methods with a list of default pre-processing methods [53]. In both categories, there are several papers that humans have been involved to improve the quality of the data set and ML performance. To exemplify, for human collaboration in the first category of the data pre-processing approaches, some studies are described as follows. To cope with missing values in data set, Ye et al. [54] proposed a Bayesian-network based missing value imputation approach that was enhanced by crowdsourcing. Crowdsourcing was introduced as an external source where ordinary people can provide contextual knowledge and use their cognitive ability to deal with missing values. In Doan et al. [55], different human-in-the-loop challenges and opportunities related to the entity matching problem in “Corleone/Falcon” project were investigated and the efficiency of the human collaboration in data integration, trust to the power of humans for better results and attention to the humans’ relations to deal with the problem when a group of humans are in the loop were reported as the most important lessons learned in this paper. In Siddiqui et al. [56], a human-guided anomaly discovery method was proposed so that humans were responsible to select actual anomalies from a ranked list of anomalies while this problem was designed within a framework of optimization.

In [53], Berti-Equille designed a method named Learn2clean to select an optimal sequence of data cleaning tasks to achieve the best results so that in case that the model was uncertain to select the cleaning tasks a human expert was responsible for that. In Chu et al. [57], a cleaning system named KATARA based on crowdsourcing and knowledge bases was presented which humans had the role of validating and verifying data when knowledge bases were not able to interpret knowledge semantics to generate possible repairs for incorrect data. In Assadi et al. [58], in addition to the emphasis on the necessity of the human expert collaboration in data cleaning, a data cleaning system by interacting with human experts (called DANCE) was presented which optimized the use of human experts for pruning errors in query results by assigning some weight to the suspicious data issues.

#### 3.2.2. Human-in-the-Loop for ML Modelling

In this section, we investigate human–AI interaction when it comes to ML modelling in three important stages of the ML processes including feature selection, model creation and model selection. In the feature selection section, we survey the role of humans in providing and selecting features. In the model creation section, human interaction in the learning process of ML methods is investigated and finally in model selection section, human collaboration in selecting an ML model among different ML approaches is studied. In Figure 2, a general framework related to human–AI interaction for ML modelling is shown and the role of humans to address ML challenges in HILML modelling is highlighted.

Feature selection

On the issue of features in ML methods, the role of humans is cooperation to generate and select features. Although providing and selecting features by humans had been replaced by classifiers in recent papers, the role of human specially when training data is limited, features are not sufficiently discriminative and features should be transformed cannot be ignored in ML applications [15,59,60]. In fact, detection and selection of informative features need domain knowledge and human inspiration for better ML outputs [61].

The role of human in feature selection can be classified into two general categories in HILML. In the first category, only humans are responsible to generate and select features while in the second category humans collaborate with automatic computer-based feature selection approaches. To exemplify, humans have the role of selecting and providing features in the following papers in the literature. In Kim et al. [62], crowd workers interactively collaborated with a Bayesian case model in feature selection while users were also asked about the degree of confidence they have in their feedback. In Correa et al. [63], an interactive textual feature selection approach was proposed so that human experts elicited a set of high-level features composed of correlated words based on their experience for consensus clustering. In Wang et al. [64], human EEG signals were used to determine more desirable features to train a generative neural network. Zou et al. [65] presented a feature discovery approach by engaging crowd workers via a comparison query (“two-out-of-three” similarity queries) to present an adaptive algorithm.

In the automatic machine learning approach, three general methods have been presented for feature selection including filter methods, wrapper methods and embedded methods [32,66]. The filter methods are related to selecting the features based on their rankings against some score functions. Wrapper methods rank features regarding some performance measures like accuracy while it requires retraining the data set. In embedded methods, feature selection is performed while the learning algorithm is in process. Different human-in-the-loop approaches have been presented for these three feature selection approaches resulted in selecting high-quality features. For example, in Hu et al. [67], a kind of filtering method for feature selection was proposed that used a cluster-based feature selection scheme. All features were ranked and a list of highly-ranked features was presented to users for labelling. Features accepted by users in the current iteration as well as highly ranked features from the previous iterations provided the next feature set for the next clustering iteration. Drobnic et al. [29] presented a wrapper method in feature selection that domain human expert knowledge was used once the problem was selecting among equally important features in the proposed approach. Correia and Lecue [32] presented a human-in-the-loop embedded feature selection method by jointly training the learning algorithms and the feature selection through gradient descent. In their paper, human experts were asked to identify the most related features for a few instances in the data set. Then using reinforcement learning, the probability of selecting each feature was derived to produce a subset of features using humans’ feedback for each instance in the data set. Cheng et al. [31] used crowdsourcing to generate features and introduced an embedded method resulted in an accurate classifier. In their paper, crowd workers were asked to provide features through repeated comparison questions and the reasons behind that so that the final classifier could improve both ML performance and human judgment particularly when automated feature extraction and selection is not feasible. Takahama et al. [61] presented an embedded feature selection method, named AdaFlock inspired by AdaBoost, by defining features and obtaining informative features iteratively using crowdsourcing based on misclassified instances.

Model creation

The knowledge, capability and domain expertise of humans are useful to improve ML models to solve the real-world problems. Human-in-the-loop provides the chance to involve humans in the learning process of ML methods and improve their performance. In this section, we classify human–AI interaction in the learning process of ML methods into three main categories including adding new information to the learning process directly, determining and modifying parameters of ML methods directly and modifying parameters of the ML methods using parameter learning based on other human interactions.

Human can collaborate with ML approaches and improve their performance by adding knowledge to the learning process. However, this kind of collaboration of human experts and ML methods is limited. For example, Constantinou et al. [68] introduced a Bayesian network (BN) which the expert knowledge about factors that are important for decision making was integrated. The expert knowledge was added to the network as a new variable and then the network was updated. Another type of human–AI interaction in this category is related to the rule-based ML methods. For example, Yang et al. [28] proposed a collaborative ML model named Rules Learner for text analysis in which domain experts could provide some rules in addition to rules derived by deep learning or update the derived rules via a user interface. The proposed model was more generalizable specially when few data is available. In Fogarty et al. [69], an interactive concept learning approach was introduced to define rules by humans and rank images according to the visual characteristics.

In HILML, human knowledge can be integrated with ML models via defining a constraint in the learning process. The most important application of this kind of human collaboration has been investigated in human-in-the-loop reinforcement learning. Reinforcement learning as a branch of ML approaches deals with using experience got through interacting with the world to take steps in the environments to improve the ability of generating high-quality decisions which are based on simple reward feedback. Human-in-the-loop reinforcement learning concerned with how humans collaborate with reinforcement learning agents to guide them towards the desired behaviour. In human-in-the-loop reinforcement learning modelling, humans have had different roles to guide agents. Four types of human knowledge integration methods in human-in-the-loop reinforcement learning are as follows [70]: reward shaping which tailors the reward function of the reinforcement learning through human feedback to customize the behaviour of the agent in order to fit it to the human intention [71], policy shaping that augments the policy of the agent and formulates the human’s feedback as action advice which is usually a binary critique [72], guided exploration process that minimizes the learning process through injecting human knowledge to guide agent exploration process to state with a higher reward [73] and augmented value function that tries to augment a value function through combining a value function created by agent and the one created by human feedback while the value function is an estimated of the expected future reward when following the current policy [74]. The most commonly used human integration strategy in this area is rewarding as a feedback signal to map human information to reward for better model control [70,72]. For example, as one of the first try in human-in-the-loop reinforcement learning literature, in [75], Isbell and Shelton proposed a human-in-the-loop reinforcement learning in which crowd workers were responsible to create the reward function. In [76], Knox and Stone investigated learning from human rewards in reinforcement learning through six experiments. In [77], Thomaz and Breazeal presented a human-in-the-loop reinforcement learning method that the role of human was giving reward signals to the learner by providing future-directed rewards as well as past-directed rewards to guide the model. Applications of human-in-the-loop reinforcement learning to deal with real-world problems have been investigated in few papers. For example, in Shah et al. [78], a human-in-the-loop reinforcement learning approach was presented for task-oriented dialogue management and in Winter et al. [79], to program an assembly task, human-in-the-loop reinforcement learning approach was used.

Direct manipulation of ML methods parameters is another human–AI interaction in ML applications. Although this kind of parametric interaction usually needs a deep understanding of the ML methods [80], humans are able to interact with ML methods specially when ML methods are transparent. Specifying weights for features and data samples, determining the number of clusters in clustering, to name but a few are some examples of this kind of collaboration. To exemplify, the following papers demonstrate human–AI interaction in direct ML parameter tuning. In [81], Kapoor et al. provided the possibility for users to change confusion matrix values to express their preference for improving classifiers’ results. In Hu et al. [82], humans were able to determine weighs for features in document clustering and cluster data according to the determined high-level features. In Arin et al. [83], human experts were able to adjust the number of clusters and also the similarity threshold parameters via an interface in clustering.

The final category related to model creation in human–AI interaction is indirect ML parameter tuning via parameter learning based on other human interactions. Visualization systems in HILML are important instruments for this kind of parameter tuning in ML applications. When humans are interacting with ML methods, they expect to adapt model parameters via an interface by observing the output results of ML methods in visualization systems. Different papers in the literature have presented different approaches in this area. For example, in Schneider [84], human experts could determine an area in a visualization system related to the classifier outputs and modify the outputs via indirect parameter tuning. In Fails and Oslen [13], humans determined some regions in images for image classification and gave a permit to the model to classify features based on the information they provided. Usually, the aim of parameter tuning (directly or indirectly) in HILML applications is ML refinement. So, although human knowledge can help to generate a powerful ML model or defining parameters at the start point of applying ML methods can improve the performance of ML methods, parameter tuning is very practical in ML refinement stage in HILML applications. In this process, human experts have the opportunity to learn how to guide model to get desired results.

Model selection

In ML, to solve real-world problems, model selection indicates procedures in which a specific ML method can be chosen among a set of candidate ML methods to be used according to different criteria which the most important ones are characteristics of the data set, features’ structure and performance of ML methods. In the literature, especially when it comes to processing complicated tasks, we can see that humans collaborate to select the ML methods regarding the type and characteristics of the data set. For example, Perry et al. [85] presented an efficient bridge inspection system implementing Unmanned Aerial Vehicles (UAVs), computer vision techniques and three clustering approaches with human–AI interaction in model selection. In their paper, to segment different parts of a bridge in its 3D point cloud model, three clustering methods were used that each one was suitable for the special shapes of the bridge elements (data samples). These clustering methods were Gaussian Mixture Model (GMM), Agglomerative Clustering and a clustering approach based on the surface normal of each point within the point cloud. It was the role of humans to see the shapes and select the related clustering method and determine the order of using clustering methods in this application.

In addition, when humans evaluate the performance of an ML method, they may consider different criteria like the accuracy of the model, time and so on and decide to change the ML approach according to their criteria. For example, in Fiebrink [86], to investigate the application of interactive supervised learning on different gestural analysis problems in computer music, researchers evaluated different criteria to assess the quality of the trained models and got the result that through interaction with learning methods users will be able to modify learning problems to generate the models that are more useful and practical in applications.

Ensemble learning as an ML approach which combines multiple classifiers to build one classifier superior that its components can provide an introduction for presenting human-in-the-loop model selection approaches in HILML applications. One of the first try in this area returns to Talbot et al. [87]. In their paper, an interactive visualization system named “EnsembleMatrix” was proposed which provided a graphical view of confusion matrixes for users so that they could investigate relative metrics of different groups of classifiers on different feature sets. So, users could select the best classifiers combination with better performance to improve the accuracy of the ensemble classifier. While “EnsembleMatrix” focused on selecting the best combination of classifiers among different combinations, in Schneider et al. [84], despite considering the combination of classifiers, the performance of each classifier was accessible and steerable by a visualization system for human experts and both data space (classification results) and model space (ensemble of classifiers results) were integrated with ensemble learning via visual analytics to help human experts for getting their desired results. In this paper, users could select classification output regions, observe how different classifiers classified the area, add and remove classifiers in an ensemble and track the performance of the different ensemble models.

#### 3.2.3. Human-in-the-Loop for ML Evaluation and Refinement

In HILML applications, the role of humans as users of the ML methods to evaluate and refine ML outputs is important because humans should be satisfied and use ML methods. Humans evaluate ML outputs and decide to refine the method or accept that. In this section, firstly, we investigate the role of human in the evaluation of ML methods, then explain the role of human in human–AI interaction for refining ML outputs. A general framework of human–AI interaction in this ML stage as well as ML challenges addressed by this interaction is shown in Figure 3.

##### Human-in-the-Loop for ML Evaluation

The role of human to evaluate ML performance is a key role that affects the applicability of ML methods. Human experts observe and evaluate the outputs of the ML methods and control the ML methods via different interactions. For example, in Liang et al. [88], ML methods were used for shooter localization using social media videos and a human evaluated and verified the ML results in each stage of the algorithm using a web interface with the aim of insurance of the accuracy of the estimations. In Alahmari et al. [89], a domain expert was responsible for accepting and rejecting outputs of two implemented ML methods so that this intervene improved the performance of the ML methods.

To evaluate the performance of ML methods, several conventional and popular metrics such as accuracy, precision, recall and so on have been used. When humans are in the loop, the evaluation of ML methods is an open challenge [11]. In general, evaluation measures found in HILML literature can be classified into two categories of objective and subjective measures. Objective measures commonly are related to the performance of the ML methods in terms of numeric measures while subjective measures are generally associated with the satisfaction of the human. In several papers, humans have used one kind of these measures to evaluate the ML performance while in other papers both kinds of measures have been applied for ML evaluation. In the following sections, different subjective and objective evaluation measures that humans have been used to evaluate ML methods are described.

Subjective measures for ML evaluation

The most important subjective measure for ML evaluation is human satisfaction. Human satisfaction indicates the essential role of the humans as the users of ML methods for ML evaluation. Several papers have considered different domain human experts with different level of expertise to evaluate the performance of ML approaches. For example, in weber et al. [6], the satisfaction of different groups of people with various characteristics and different levels of expertise when they were using the proposed ML method was considered as a measure to evaluate the performance of the ML method. In Yuksel et al. [34], the satisfaction of blinded users from the ML approach to producing video description was introduced as an evaluation measure for the performance of the proposed HILML approach and evaluation results verified the outperformance of HILML than only ML approach from the view of users. In Kapoor et al. [81], desired performance of the classifiers based on human wishes was considered as a criterion for ML evaluation and refinement.

Objective measures for ML evaluation

Different numerical measures like accuracy, recall, F-score, precision, posterior proximities, likelihood and so on that are used to evaluate ML methods, have been applied by human experts to assess the ML methods outputs in HILML for ML evaluation. In this regard, humans usually determine a threshold for numerical measures and evaluate the performance of the ML methods through it. For example, In Michalopoulos et al. [42], an F1-score threshold was determined by humans to accept the ML results, otherwise, the ML method should be refined to get desired F1-score value. Also, F-Score was the evaluation measure used in Kwan et al. [90] to test the performance of the presented HILML approach in tagging biomedical texts in text mining.

##### Human-in-the-Loop for ML Refinement

Human–AI interaction with the aim of ML refinement is the key task in HILML literature. In fact, in ML applications, humans try to steer and guide ML methods to get desirable results. According to the recent research, by explaining the prediction of ML methods to the domain human experts, they will be able to refine the outputs by providing essential correction back to the method [91]. In different stages of human–AI interaction including data producing and pre-processing, model creation and model selection that described in the previous sections, humans can interact with ML methods to refine ML outputs. In data producing stage, adding new instances to the training data set and labelling or relabelling some instances of the training data set are strategies that humans use to guide the ML method and refine ML outputs. In model creation, parameter tuning is the most import stage that humans can steer ML results toward the desired outputs. Also, in model selection, humans can refine ML outputs by selecting different ML methods regarding the characteristics of the data set, features and research area to get acceptable ML results. As in previous sections we described these strategies related to human–AI collaboration to refine the ML outputs, in this section we explain how human interact with ML methods to refine ML outputs in terms of the number of interactions.

Human–AI interaction for ML refinement to get desired outputs

When the final outputs of ML methods are not satisfactory for the users, they can interact with ML methods iteratively till they get their desired results. Several examples of this kind of human–AI interaction have been presented in HILML applications. For example, In Yang et al. [28], human expert interacted with the ML model iteratively until his desired results in terms of the F1 measure, precision and recall be achieved. In Weber et al. [6], domain human experts collaborated in the learning process by providing new data samples iteratively to refine ML outputs in image reconstruction application. In Michalopoulos et al. [42], human expert collaborated with the ML method by providing data labels interactively until a desirable result be achieved.

Human–AI interaction for ML refinement at specific times

The collaboration of humans with ML methods for ML outputs refinement is terminated at specific times in several papers and usually this repetition number is determined as the number of iterations by the users. For example, in Tkahama et al. [61], the number of experiments for collaborating humans with the ML method was specified by researchers. Also, In Kwon et al. [90], the role of human to refine the ML results was tested in five iterations and this number of iterations was determined by humans to get acceptable results.

Human–AI interaction for ML refinement once

In some ML applications, humans interact only once to refine ML outputs. Some examples of this kind of human–AI interaction are as follows. In Yuksel et al. [34], human experts refined the ML produced video description for blinded people once at the end of the ML process. In Bahrami and Chen [24], the human was in the loop for ML results refinement once where the ML method was not able in information extraction from web API documentation.

### 3.3. Who Are the Humans in the Loop?

In HILML, it is dispensable to leverage domain human expertise in ML methods, so, human experts need not possess a deep understanding of ML methods and domain expertise can improve the ML understanding of the concept being trained [15]. Depending on the ML method, application area, the role of humans in the loop and the level of task complexity, different levels of domain expertise are essential for humans to be in the loop. In some applications, especially in medical applications, the level of domain expertise is higher for humans who are interacting with ML methods. Also, at the tasks with the higher complexity in which humans interact with the ML methods such as refining the classifiers, feature selection and other intelligent functionality, more domain expertise should be applied than issues related to lower complex tasks like selecting examples which humans like and do not like [92].

Intrinsic abilities of humans are the only expertise that humans should have in some HILML applications. Using the wisdom of crowds or crowdsourcing as an interesting topic to increase the efficiency of ML methods can be considered in this category [17]. Although due to the uncertain quality of crowd workers’ performance the results of the cooperation between ML methods and humans are sometimes unreliable, the good performance of this kind of HILML approaches is reported in the literature. Liu et al. [22] implemented Amazon Mechanical Turk for crowdsourcing to use the intrinsic ability of people in pattern recognition. In Sakata et al. [23], crowd workers were responsible to provide data for Neural Network. In this paper, although the humans should have the expertise to determine the values of the features, this shortage was compensated to some extent by designing some simple questions could be answered by crowd workers (using intrinsic human cognition capabilities).

In addition to the crowdsourcing approach, the ordinary abilities of humans are enough for humans to be in the loop in some HILML applications. So, a high level of expertise is not needed for this kind of ML applications. For example, in Perry et al. [85], humans or users who were in the loop should have basic knowledge in the related field. In Feder et al. [25], the ability to identify demographic traits was enough for a human to be in the loop. In Wen et al. [26], as humans can determine a moving target in a video stream without the need for any special expertise, no special expertise was essential for the humans to be in the loop.

In HILML applications, when the task is professional and complicated, we observe that humans with a higher level of domain expertise should be in the loop. For example, in Weber et al. [6], the human should have expertise in the field of image restoration to help the ML to get desirable results. In Roccetti et al. [4], human experts who defined some semantics to select high-quality data sets had expertise in the application area. In Wrede et al. [48], the human who was responsible for labelling should have domain expertise in the field of generating gene network modelling. In Alahmari et al. [89], the human who had the role of the evaluator of the ML outputs should have domain expertise in the field of stereology to recognize acceptable outputs. In Yang et al. [28], the human who was responsible for model creation and refinement should have domain knowledge and expertise to help the ML method for better text classification.

Generally, for the humans who collaborate with ML methods, the more the level of the expertise is, the better human–AI interaction outputs are achieved. However, because of some additional subjects like cost and time, domain human experts are replaced by crowd workers or humans with low level of expertise in some ML applications. Several papers have compared the performance of humans with different levels of expertise in HILML applications and verified the importance of the domain expertise for humans who are in the loop. For example, in Snow et al. [46], the quality of labels provided by crowd workers and domain expert labellers was compared to be used in natural language processing and results showed the better performance of domain experts than non-experts in data labelling.

### 3.4. How Do Humans Interact with ML in HILML?

In this section, we present a general framework for human collaboration in HILML applications according to the number of human interactions in this process. In this framework, we consider all stages of the ML processes for human–AI interaction in ML applications. In Table 1, a summarized description of human–AI interaction in terms of the number of interactions is shown.

Humans iteratively interact with ML methods

In some ML applications, humans collaborate with ML methods iteratively. The ML method gets information from the human, uses it and provides new outputs, human provides information again, gives it to the ML method and this process will be continued until the stopping conditions be satisfied. These stopping conditions can be the satisfaction of the user, no improvement in the ML results, the achievement of a determined objective measure and so forth. To exemplify, in Alahmari et al. [89], human expert was iteratively in the loop of the proposed ML approach to evaluate the performance of the ML method and decrease the error of applying the proposed ML on test data set. In Yang et al. [28], humans were iteratively in the loop to get their acceptable results.

Humans interact with ML methods at specific times

Depending on ML applications, humans interact with ML methods at specific times. For example, In Perry et al. [85], we can see that human interacted only twice with the ML method. At the first time, human selected the initial point of a clustering method and in the second time, the human was responsible for model selection. In Feder et al. [25], human collaborated twice with a classifier by relabelling some instances. However, testing more collaboration iterations was proposed for future work. In Wen et al. [26], human determined the labels for some instances in the data set at the first step only once and in continue if ML methods had some errors, human relabelled some instances to refine the ML outputs.

Humans interact with ML methods once

In several papers, humans interact with ML methods only once in applications. To exemplify, in Rocettin et al. [4], human experts interacted with ML once by defining some semantics to select the high-quality data set instances in order to improve the prediction power of the neural network. In Netzer and Geva [47], human experts collaborated with ML process once for data labelling to tag target visual objects in images. In Yuksel et al. [34], human refined the ML results once at the end of producing video description for blinded people.

### 3.5. Human-in-the-Loop in Medical ML Applications

Medical ML is one of the most significant concerns for data mining community. Generally, applications of ML methods in medicine have been investigated in four different branches including bioinformatics (medicine at the molecular and cellular level), imaging informatics (medicine at tissue and organ level), clinical informatics (medicine at single patient level) and public health informatics (medicine at the society level) [93]. Although ML methods especially when a lot of data samples exist have had a remarkable performance in prediction and pattern recognition, the performance of ML approaches is not unanimously approved by medical experts [94]. This issue encourages data mining community researchers to present new approaches which can be accepted and implemented by medical experts. HILML can be one solution for this issue to implement the power of both ML methods and domain human expertise to achieve acceptable and desirable prediction outputs.

When it comes to medical ML applications, medical experts need to be able to understand, validate, refine and trust ML methods [95]. Understanding the reasons behind a decision made by ML methods is the first prerequisite for the next decisions in medicine [28]. ML transparency and interpretability that can be generated in HILML approach by the collaboration of medical experts and ML even when ML methods are black-box provide this ground for medical experts to use ML approaches for prediction.

The performance of ML methods that is measured specially by accuracy, quality of ML results as well as subjective matters about the ML outputs has been introduced as issues that medical experts express as a barrier to use ML methods in medical applications [94]. Using human expertise accompany with providing an interface for human–AI interaction enable medical experts as the users of ML methods to meet their needs when they are using ML methods. Medical experts can see the results, validate them and refine them to get their acceptable outputs. This process improves medical experts’ trust in ML methods.

In this section, we describe the role of medical experts in medical HILML applications regarding the framework presented in Section 3.2 to discuss the potential of HILML to meet medical experts’ expectations. Also, we raise future research opportunities in this research area. In Table 2, human–AI interaction in medical ML applications regarding different stages of HILML approach are shown.

#### 3.5.1. HILML for Data Producing and Pre-Processing in Medical Applications

Data quality has a significant role in the quality of ML outputs. In HILML, using domain human experts for data producing and pre-processing can help to generate high-quality training data set for better ML prediction. In this regards, data producing and pre-processing in different scopes of medical application fields need domain expertise and generally cannot be performed by ordinary people and crowd workers due to subjects like the quality of the samples and labels, subject concepts, privacy and so forth. [89].

As mentioned before, producing whole data set, add some new samples to the data set, providing labels for whole samples in the data set and providing labels or new labels for some samples in the data set are human–AI interaction steps in HILML for data producing. Generating data set samples and data labelling before ML training process in medical applications can be carried out by human experts according to the application area and the characteristics of the under-investigation case study. Medical experts’ collaboration in this process can result in providing high-quality training data set. However, this process is very time consuming and expensive. For example, in Huang et al. [41], medical experts were responsible to generate the whole data set by scoring the features of breast tumours in ultrasounds for better tumour diagnosis. In Yimam et al. [96], medical experts annotated or labelled data to develop a biomedical entity recognition data set with the collaboration of an ML approach.

Despite the effective role of medical experts in generating high-quality training data set before ML training step, the presence of a medical expert with the high level of expertise is more necessary during ML training. To refine ML outputs, we need medical experts to be able to interpret initial ML results and guide ML process toward desired results by adding new samples to the training data set and labelling or relabelling some samples. To exemplify, in Wrede et al. [48], human experts had the role of data labelling in exploring gene regularity network models to inform the model and guide it toward interesting behaviours. In Alahmari et al. [89], to improve the segmentation and counting of cells on Extended Depth of Field (EDF) images, a HILML approach was presented in two stages. In the first stage, an unsupervised ML approach named Adaptive Segmentation Algorithm (ASA) was used to provide initial labels for training data set. In the second stage, the labels were evaluated by medical experts and only samples with accepted labels used as training data samples of the proposed deep learning method. The samples that were rejected by human experts were used as test data for deep learning and if the prediction results (labels) were acceptable by the human experts, the samples were added to the training data set of deep learning approach iteratively.

Active learning in medical ML applications verifies the importance of medical experts’ collaboration in data labelling to reduce the computational cost and improve the performance of ML methods. Recently, active learning-based approaches can be observed more in medical HILML applications especially in medical image analysis. For example, in Liu et al. [97], with the aim of improving training efficiency for a deep network-based lung nodule detection framework in computer-aided diagnosis, an active learning approach was proposed that a radiologist was responsible to annotate selected informative CT scans to be used as training data set for the proposed nodule detector framework. In Sheng et al. [98], to construct a knowledge graph in the medical domain, active learning was used for reducing medical experts’ interaction cost and controlling the quality of the medical knowledge graph. Active learning is an effective approach to implement medical experts efficiently in data labelling and in future work of medical HILML applications this approach can be a research opportunity in different fields of medicine.

The importance of data pre-processing in medical applications to improve data quality can increase medical experts’ collaboration with ML methods. Although different human-in-the-loop approaches have been proposed in the literature to deal with various data errors consist of missing values, outliers, noisy data and so on in different research areas, despite different anomalies seen in medical data sets, human–AI interaction in the medical application can be a research opportunity for the data scientist to use medical expertise and present different HILML approaches for medical data pre-processing regarding types and characteristics of medical data and case studies.

#### 3.5.2. HILML for ML Modelling in Medical Applications

Human–AI interaction in medical ML applications with respect to ML modelling can take place in the stages of feature selection, model creation and model selection. The collaboration of medical experts with ML methods in providing and selecting features is critical in medical HILML. Applications of different feature selection approaches in medicine have been investigated in various papers. However, the presence of the medical experts in this area is very faint and this provides an opportunity for researchers to present HILML approaches in feature selection to improve the performance of the ML prediction in medical ML applications. Human–AI interaction in medical ML applications for feature selection is more associated with medical imaging. To exemplify, in J. Cai et al. [99], it is described that medical images contain different features in medical ML applications and medical experts have knowledge and experience to select the features that are diagnostically important from case to case. In conclusion, it seems that the collaboration of medical experts for feature selection in different medical ML applications can leverage medical knowledge and experience to ML approaches and this research area needs more studies in medical HILML applications. To review the feature selection approaches in medicine, researchers are referenced to Remeseiro and Bolon-Canedo [100].

Human–AI collaboration in medical ML applications concerning model creation steps can be seen in parameter tuning and adding medical experts’ knowledge to the learning process. For parameter tuning in medical applications, indirect manipulation of parameters by visualizations systems is an applicable and interesting human–AI interaction area to involve human knowledge and experience in the ML process. In fact, as direct parameter tuning in HILML needs data mining knowledge, human–AI interaction via indirect parameter tuning in medical ML applications has suggested in recent papers. So, designing an accurate and suitable ML user interface has an important role in the success of medical HILML applications for model creation. For example, in J. Cai et al. [99], by designing a user interface for medical decision making, medical experts could retrieve visually similar medical images from other patients to decide for a new patient. In this process, medical experts could change parameters of a deep learning approach indirectly by cropping regions in images for emphasis on the importance of that region to find similar images and selecting some images they thought were more related to the concept they were looking for. Indirect parameter tuning in medical ML application via human–AI interaction can be hot research opportunity in HILML applications.

Human–AI interaction in medical ML applications by adding medical expert’s knowledge as rules or constraints to the ML process can improve ML performance and increase medical experts’ satisfaction. In rule-based ML approaches, medical experts can provide some rules to add to the ML process specially when data sets cannot capture all assumptions about the under-investigation case study. In future research, developing new rule-based ML approaches by collaborating with medical experts can increase the application of ML methods in medicine. In human–AI interaction for ML modelling, the human can also add their knowledge in the ML process by adding some constraints to the model. Human-in-the-loop reinforcement learning in medical ML applications provides this ground for medical experts to interact with ML methods and steer them to achieve desirable results. So, human-in-the-loop reinforcement learning can be a research opportunity in medical HILML applications to use the power of medical expertise and ML methods for better prediction in future research.

In model selection, human–AI interaction can create an interactive ML evaluation and refinement process via a user interface to improve the prediction accuracy in ML applications. Using ensemble learning approaches in medical applications can give this chance to medical experts to select the best combination of ML methods for prediction. Human-in-the-loop ensemble learning can be a research opportunity in this area for future research.

#### 3.5.3. HILML for ML Evaluation and Refinement in Medical Applications

In medical ML, medical experts are users of the ML methods and usually evaluation criteria are defined and determined by them in medical HILML. Although objective measures are used for evaluation of ML approaches in medical applications, subjective measures and medical experts’ satisfaction are important for ML outputs assessment. For example, in Cai et al. [99], the satisfaction of 10 pathologists as users of the proposed HILML approach was defined as a measure to evaluate the presented approach.

The key role of human experts in HILML approaches is refining ML outputs. Regarding previous sections, medical experts can interact with ML methods and refine ML outputs regarding different evaluation measures. According to HILML and medical ML application literature, it seems medical experts prefer to interact with ML methods iteratively and refine ML outputs. For example, in Alahmari et al. [89], experts were responsible to check the ML outputs that were the types of cells in EDF images iteratively to increase the prediction accuracy of the proposed human-in-the-loop deep learning. This kind of human–AI interaction in medical ML applications can improve ML performance, increase intelligibility and explicability of ML methods and teach both medical experts and ML methods. So, for future research, our suggestion is designing this kind of human–AI interaction approaches for ML refinement in medical applications.

## 4. Conclusions

Thanks to the power of AI, different ML methods have been created to successfully deal with real-world problems. However, ML methods have some drawbacks and do not always provide satisfaction. For example, ML methods are not transparent, ML outputs are not always accurate and trustable in complicated and sensitive case studies and ML methods cannot cover all assumptions and conditions of the problems.

Humans as the users of the ML methods can interact with ML methods and improve ML performance with their knowledge, experience and expertise. This interaction between domain human experts and ML methods in ML applications has generated a new term in ML applications named HILML. HILML enables domain human experts to collaborate with the ML methods so that they can engage their knowledge, experience and skills in ML process to get better ML results. Our review has provided a human–AI interaction pipeline to investigate different aspects of the human involvement in ML applications.

As the first step, we extracted three important reasons about the necessity of human involvement in ML applications including improving ML outputs quality, increasing ML transparency and explicability and satisfying subjective matters of domain human experts. Then, we investigated the role of the human in collaboration with ML methods regarding all steps of the ML process. In this step, we divided the stages of the human–AI interaction into three classifications related to the ML process named data producing and pre-processing, ML modelling and ML evaluation and refinement. In each stage, different types of human–AI interaction according to the recently presented papers in HILML applications were described and the advantages of humans’ collaboration with ML methods were explained. Moreover, we investigated the characteristics of the humans in HILML in terms of the level of humans’ expertise and classified that into domain experts and non-expert humans (crowd workers). Finally, we presented a categorization of the number of human interactions in HILML and described them.

As this paper is the first paper that investigates human–AI interaction from the viewpoint of ML stages, it can be a guide for researchers when they want to study different strategies to involve human in ML applications to improve the ML results. According to this review, it seems that designing HILML approaches can improve the power of ML methods specially to deal with complicated real-world tasks and use the knowledge, ability and expertise of humans to get desirable results.

Among different ML application areas, the HILML approach is particularly applicable in medicine because medical experts have high levels of expertise and high ability in prediction while medical data sets usually cannot cover all conditions of the case studies. So, the collaboration between medical experts and ML methods can improve ML performance in medical ML applications. However, there are relatively few studies on HILML in medical applications. In this paper, we investigated human–AI interaction in medical ML applications regarding the proposed HILML framework and presented a discussion on future research opportunities in this research area. According to this review, we concluded that HILML approach can increase the transparency of the ML methods in medicine so that both powers of ML methods and medical experts result in accurate and trustable decision making.

## Figures and Tables

**Figure 1 ijerph-18-02121-f001:**
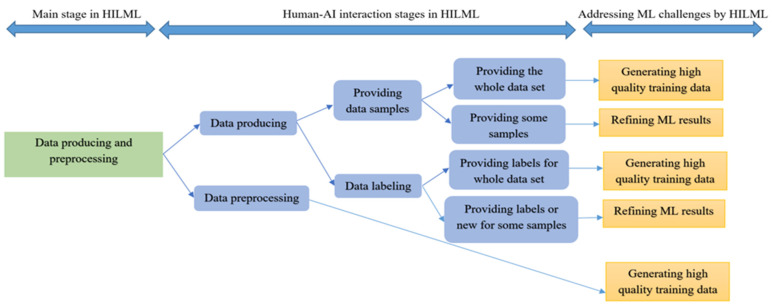
A framework overview of human–AI interaction in data producing and pre-processing.

**Figure 2 ijerph-18-02121-f002:**
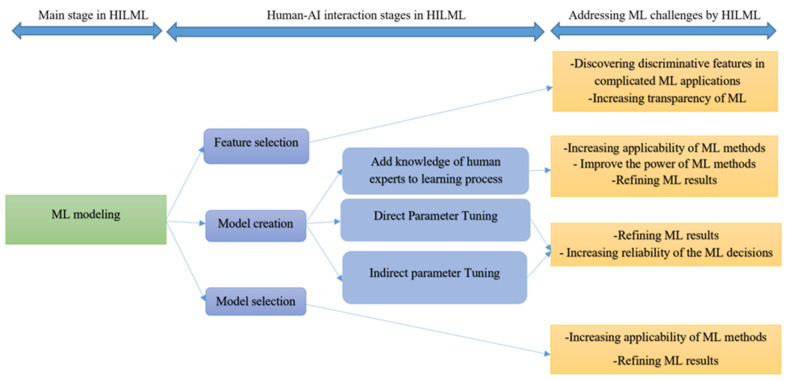
A framework overview of human–AI interaction in ML modelling.

**Figure 3 ijerph-18-02121-f003:**
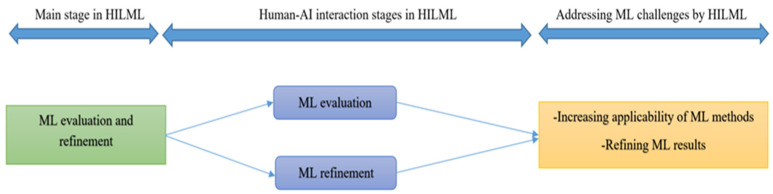
A framework overview of human–AI interaction in ML evaluation and refinement.

**Table 1 ijerph-18-02121-t001:** How humans interact with ML in Human-In-the-Loop ML (HILML).

Type of Interaction in HILML	Main Stopping Conditions	Samples of Studies
Human iteratively interacts with ML	The satisfaction of the user, no improvement in the ML results, the achievement of a determined objective measure	Alahmari et al. [89], Yang et al. [28]
Human interacts with ML at specific times	limited number of interactions is usually determined by humans as the users of the case study	Perry et al. [85], Feder et al. [25], Wen et al. [26]
Human interacts with ML once	Interaction happens once in data producing and pre-processing, ML modelling and ML refinement	Roccetti et al. [4], Netzer and Geva [47], Yuksel et al. [34]

**Table 2 ijerph-18-02121-t002:** Human–AI interaction in medical ML applications.

HILML Stage	Human Expert’ Task According to the Literature	Sample of Papers	Suggestions for Future Research
Data producing and pre-processing	Generating the whole data set	Huang et al. [41]	Active learning (Liu et al. [97], Sheng et al. [98] Human–AI interaction for data pre-processing
	Data labelling	Yimam et al. [96], Wrede et al. [48]	
	Selecting samples	Alahmari et al. [89]	
ML modelling	Feature selection	J. Cai et al. [99]	Indirect parameter tuning Developing rule-based ML approaches Human-in-the-loop reinforcement learning
	Direct Parameter tuning	J. Cai et al. [99]	
ML evaluation and refinement	Evaluation of the ML outputs	J. Cai et al. [99]	Using human experts’ criteria to evaluate and refine HILML outputs to increase the explicability of ML methods
	ML outputs refinement	Alahmari et al. [89]	

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
