# Peer review of "A Review on Human–AI Interaction in Machine Learning and Insights for Medical Applications"

_ijerph, 2021, doi:10.3390/ijerph18042121_

Round 1
Reviewer 1 Report
Comments and tips
Comments
The article is correctly written and gives an accurate analysis of the advantages and limitations of the Machine Learning methods. It also describes how human interaction has a positive effect when combined with artificial intelligence, thanks to the Human-In-The-Loop Machine Learning (HILML) method.
In particular, it points out the need for the collaboration of Medical Experts with applications that exploit Machine Learning methods in Medical Applications that use Machine Learning.
The work is well structured, with a detailed description both in the introduction and in the methodological part.
In the introduction, the attention is focused on the developments of Machine Learning and the HILML Method, defining how Human and Machine Learning can collaborate to improve the performance and to obtain more satisfactory results.
In the methodological part various aspects have been described in detail, in particular, where human-AI interaction occurs in ML Process.
This work investigates where human can be in the loop and classify the human-AI interaction process into three general categories of 1- human-AI interaction in data producing and preprocessing, 2-human-AI interaction in ML modeling and 3- human-AI interaction in ML evaluation and refinement, debating how the human interaction improve the ML performance.
Tips
In line 32 and line 33 Machine Learning methods are described. The description is very brief and it does not explain well the importance and the difference of an Artificial Intelligence software compared to a normal automatic software. I suggest you expand this part by describing it in more detail (for example by explaining the various types of Machine learning like supervised, unsupervised etc.).
Lines 49 to 50 describe how Machine Learning Methods do not have satisfactory results, but it is not explained why. I would suggest you insert some reasons with some examples that highlight the critical issues.
Lines 219 to 220 and lines 308 to 310 describe that Machine Learning Methods are unsatisfactory.
I would ask you if in this part you can give examples (e.g using sample data) and feedbacks about the performance evaluation, to show that these ML methods are not satisfactory.
In chapter 5, from line 854 I would recommend, if possible, to insert a table after line 884 highlighting the characteristics of the three methods (Human iteratively interacts with ML methods, Human interacts with ML methods in limited times, Human interacts with ML methods ounces). This makes it easier to understand the differences.
Finally, in chapter 6 from line 885, I would advise you to insert the table after line 1040 with the three applications of HILML (HILML for data producing and preprocessing in medical applications, HILML for ML modeling in medical applications and HILML for ML evaluation and refinement in medical applications).
Author Response
Dear Reviewer 1
Thank you for your useful comments. I edited the paper according to your comments which is highlighted in yellow in the manuscript.

Reviewer 2 Report
The manuscript falls within the scope of the journal. Overall, the manuscript is well-designed and quite readable. However, it seems to be unclear along several lines making the technical aspects appear weak. Major revisions are recommended.
1) This manuscript requires a systematic revision using "Guide for Author" because the abstract, introduction, methods, review results, and discussion/conclusion are not logically linked.
2) What's the purpose of this study? It is necessary to explain how this review article is needed and how it can contribute.
3) Abstract needs to be clearly outlined. The abstract should follow the style of structured abstracts.
4) The process of deriving the review results needs to be explained in Methods.
5) Figures require corrections. They are not legible, and their quality leaves much to be desired.
6) The review contents should be made more clear and concise using Tables.
7) Discussion requires a substantial rewrite to reflect the findings better. Also, the usefulness and implications of the research should be included.
Author Response
Dear Reviewer 2
Thank you for your useful comments. I revised the paper according to your comments. please see the attachment.

Reviewer 3 Report
This manuscript reviews the major work in HILML applications and investigated human-AI interaction in three main stages related to ML approach including data producing and preprocessing, ML modeling, and ML evaluation and refinement. In addition, the latest papers in the medical field are reviewed and open opportunities for future research as HILML are discussed.
This article provides the reader with insight into the pipeline of human-AI interactions. The authors even provide the first comprehensive summary of current research on human-AI interaction to identify the role of humans in collaborating with ML methods regarding all steps of the ML process. In addition, the motivation and purpose of the study are clearly explained, making it easy to grasp the presentation and focus of the article.
Author Response
Dear Reviewer
thank you for your time and notes.
Reviewer 4 Report
The paper deals with a very interesting topic, which is the role of human-AI interaction to improve the results of ml-based approaches.
The number of papers and of researches reported is remarkable and the paper is of great interest for all researchers dealing with ml, ai, and human-in-the-loop.
However, some aspects need improvement:
1) on p. 4, line 183 et seq. the authors say that they "present a framework for human-AI collaboration." It is not clear what the framework is and how it has been built: the authors should provide more details, even technical, about the framework
2) the paper is wordy, the concepts are repeated and unclear ->authors should be more concise, make the concepts more clearly visible, also with better formatting of the text (subparagraphs, bulleted lists, ...)
3) why do the authors consider only medicine as a field to be investigated in more detail? Other areas could certainly be investigated, e.g. engineering, etc...
Author Response
Dear Reviewer 4
Thank you for your comments. we revised the paper according to your comments. please see the attachments.

Round 2
Reviewer 2 Report
Thank you very much for your revised version of the manuscript. The article is well revised, reflecting the reviewer’s comments. I think the paper meets the standard of the journal, and the revised manuscript is now acceptable.
Reviewer 4 Report
The paper has been revised and now it looks to be more readable. I suggest accepting it in its present form.